# Effects of Electrolytes on the Electrochemical Impedance Properties of NiPcMWCNTs-Modified Glassy Carbon Electrode

**DOI:** 10.3390/nano12111876

**Published:** 2022-05-30

**Authors:** Sheriff A. Balogun, Omolola E. Fayemi

**Affiliations:** 1Department of Chemistry, Faculty of Natural and Agricultural Sciences, Mafikeng Campus, North-West University, Private Bag X2046, Mmabatho 2735, South Africa; sheriffawedabalogun@gmail.com; 2Material Science Innovation and Modelling (MaSIM) Research Focus Area, Faculty of Natural and Agricultural Sciences, Mafikeng Campus, North-West University, Private Bag X2046, Mmabatho 2735, South Africa

**Keywords:** supercapacitor, nickel phthalocyanine, multiwalled carbon nanotubes, bromate, electrochemical impedance spectroscopy

## Abstract

The supercapacitive properties of synthesized nickel phthalocyanine multiwalled carbon nanotubes nanocomposite on a glassy carbon electrode (NiPcMWCNTs-GCE) in four different electrolytes were investigated. The successful synthesis of the NiPcMWCNTs nanocomposite was confirmed by UV/vis electrode spectroscopy, SEM, TEM, EDX, and XRD techniques. The supercapacitive behaviors of the modified electrodes were examined in PBS, H_2_SO_4_, Na_2_SO_4_, and SAB electrolytes via CV and EIS techniques. The highest specific capacitance of 6.80 F g^−1^ was achieved for the GCE-NiPcMWCNTs electrode in 5 mM [Fe(CN)_6_]^4−/3−^ prepared in 0.1 M PBS (pH 7). Charge transfer resistance R_ct_ values of 0.06, 0.36, 0.61, and 1.98 kΩ were obtained for the GCE-NiPcMWCNTs in H_2_SO_4_, SAB, Na_2_SO_4_, and PBS electrolytes, respectively. Power density values, otherwise known as the “knee” frequency f°, of 21.2, 6.87, 2.22, and 1.68 Hz were also obtained for GCE-NiPcMWCNTs in H_2_SO_4_, Na_2_SO_4_, PBS, and SAB electrolytes, respectively. GCE-NiPcMWCNTs demonstrated the fastest electron transport capability and the highest power density in H_2_SO_4_ compared to the other electrolytes. Hence, GCE-NiPcMWCNTs-modified electrodes had high stability, high energy and power densities, and a large specific capacitance.

## 1. Introduction

In recent years, the quest for environmentally sustainable energy sources other than crude oil has prompted an upsurge in energy and power research. Supercapacitors have recently received increased attention as a potential replacement for batteries and fuel cells due to their high power density and quick charging and discharging capabilities [1]. Supercapacitors are energy storage devices with a high capacity and low internal resistance that can store and distribute energy at higher rates than batteries due to the energy storage method, which comprises a simple charge separation at the electrode–electrolyte interface. Capacitors are classified into two types: pseudocapacitors and double-layer capacitors. Non-faradaic processes are used to store charge in double-layer capacitors, whereas faradaic processes are used in pseudocapacitors.

According to the research, electrodes modified with CNT-MO nanocomposite showed a large capacitive current in several electrolytes [2,3]. Hence, determining the charge storage properties of these materials as possible energy sources is crucial. This is in reaction to the growing need for clean energy solutions, with supercapacitors proving to be the most promising energy storage and power output technology for portable devices, electric cars, and renewable energy systems [4]. Carbon nanotubes have recently attracted much attention due to their unique features and potential usage in various areas of nanotechnology. The unique qualities of multiwalled carbon nanotubes (MWCNTs), such as low resistivity, wide surface area, excellent chemical stability, and the narrow distribution of mesopore diameters, have made them viable electrode materials for electrochemical supercapacitors in the recent decade [5,6]. MWCNTs’ unique properties enable them to be employed in various applications, including electrochemical sensors and analytical, energy, material, electronic and optoelectronic, pharmaceutical, catalytic, and biomedical fields. MWCNTs have been intensively researched as modifying agents for electrochemical sensing applications due to their ease of adsorption on a bare electrode surface [7]. Notably, the properties of small dimensions, outstanding biocompatibility, high conductivity, modifiable sidewall, functional surfaces, and high reactivity make MWCNTs highly suitable materials for fabricating electrochemical sensors with extraordinary performance [8,9]. The synthesis of magnetic nanoparticles of Ni, Co, and Fe has gained more attention in recent years owing to their superior magnetic properties and impending uses in many fields, including sensors, memory storage devices, and catalysis [10].

Phthalocyanines are large aromatic macrocyclic organic compounds with nitrogen atoms linked with four isoindole units and a ring system consisting of 18 π-electrons in a two-dimensional geometry. The presence of N-4 macrocycles plays a crucial role in sensor design due to their high catalytic activity, thermal stability, biocompatibility, and chemical inertness. Phthalocyanines can act as electrode modifiers in catalyzing the electrochemical behaviors of various species due to the increased number of catalytic sites on the electrode surface [11]. The changing technological advancement of the world has recently directed a lot of attention to the development of new environmentally benign and readily accessible catalysts. Metal phthalocyanine (MPc) as a catalyst exhibits exceptional performance, which is attributed to its physical and chemical properties and symmetric macro ring [12]. Metal phthalocyanines have received enormous usage as electrochemical sensors owing to their thermal and chemical stability, ability to produce unique and optimized thin films with varying degrees of sensitivity and high selectivity, ability to change the substituents in the side chain, and ability to incorporate about 70 different metal atoms into their rings [13,14]. The interface between the phthalocyanine’s π-system and the metal’s orbitals facilitates electron transfer from the metal to the Pc ligand and vice versa, favoring central metal ion adsorption, axial ligand addition, and complex oxidation and reduction [15]. Additionally, they have also found broader applications in other fields, such as catalysts, molecular electronics, photonics, etc., due to their versatile optical and electrical properties and high electron transfer abilities [16,17,18]. Nickel phthalocyanine (NiPc), in particular, has gained popularity due to its large theoretical specific capacitance (2573 F g^−1^), environmental friendliness, and well-defined redox behavior, allowing its utilization in a wide range of applications, including supercapacitors and solar cells [19].

Nickel phthalocyanine-doped MWCNTs have recently been investigated and found to display enhanced capacitive behavior owing to the greater surface area of the MWCNTs and phthalocyanine, as well as the high stability and conductivity of nickel and the MWCNTs [20]. The best electrochemical performance of carbon nanotubes is achieved by maintaining a high surface area and, at the same time, depositing an ultrathin layer of highly conformal pseudocapacitance materials [21,22]. Hence, the best features of NiPc and MWCNTs are joined together to fabricate a hybrid supercapacitor with huge capacitance, higher energy density, and remarkable cycling stability. This work investigated the comparative supercapacitive behaviors of nickel (II) phthalocyanine multiwalled carbon nanotubes–modified GCE electrodes (GCE-NiPcMWCNTs) in four different bromate solutions of PBS, H_2_SO_4_, Na_2_SO_4,_ and SAB electrolytes via the electrochemical impedance spectroscopy EIS technique. There has not been any research on the supercapacitive properties of GCE-NiPcMWCNTs in bromate solutions of PBS, H_2_SO_4_, Na_2_SO_4_, or SAB electrolytes as far as we know. Therefore, this research compares the supercapacitive behaviors of GCE-NiPcMWCNTs in four distinct bromate electrolyte solutions for the first time.

## 2. Experimental Procedures

### 2.1. Materials and Reagents

Multiwalled carbon nanotubes (MWCNTs) (O.D. = 10 nm ± 1 nm, I.D. = 4.5 nm ± 0.5 nm, L = 3–6 μm) with 98%, potassium bromate (KBrO_3_), ethylene glycol, *N*,*N*-dimethyl formamide (DMF), and 5 mM potassium hexacyanoferrate (III)/(IV) K_3/4_[Fe(CN)_6_] solution in 0.1 M phosphate buffer solution (PBS) (pH 7) prepared from disodium hydrogen phosphate dehydrate (Na_2_HPO_4_·2H_2_O) and sodium dihydrogen phosphate dihydrate (NaH_2_PO_4_·2H_2_O) were purchased as analytical grade reagents from Sigma-Aldrich (Darmstadt, Germany). Nickel chloride, sodium acetate, sulfuric acid, sodium sulfate, methanol, and hydrazine hydrate were obtained from Glassworld Chemicals (Johannesburg, South Africa), while the hydrochloric acid (HCl) and sodium hydroxide (NaOH) used for adjusting the pH were purchased from Merck Chemicals (PTY) LTD (Germiston, South Africa). The experiment was carried out using distilled water throughout.

### 2.2. Apparatus and Equipment

The modified electrodes were morphologically and structurally characterized by transmission electron microscopy (TEM) with JEOL2100 instrument fitted with a LaB 6 electron gun by JEOL Ltd, Tokyo, Japan, X-ray diffraction spectrophotometry (XRD) by Bruker Company, Karlsruhe, Germany, scanning electron microscopy by JEOL JSM-6610 LV, Dearborn, Peabody, MA, USA (SEM), UV–visible spectrophotometry by Agilent Technology, Cary 300 series UV−vis spectrometer, Darmstadt, Germany, and Energy diffraction X-ray (EDX) by JEOL JSM-6610 LV, Dearborn, Peabody, MA, USA (EDX). Both electrochemical and impedance spectroscopy (EIS) studies were conducted using an Autolab Potentiostat PGSTAT 302 (Eco Chemie, Utrecht, The Netherlands), controlled by GPES (version 4.9) software (Utrecht, The Netherlands). At the same time, the EIS data were obtained using Metrohm Autolab (FRA 32) NOVA 2.1.3 software (Utrecht, The Netherlands) with a frequency range of 100 kHz–0.1 Hz.

### 2.3. Nickel Nanoparticle Synthesis

Nickel nanoparticles (NiNp) were made by completely dissolving 5.0 mL of hydrazine hydrate and 0.955 g of nickel chloride in 400.0 mL of ethylene glycol, and 4.0 mL of 1 M NaOH solution was added to the reaction mixture and swirled at 60 °C for 1 h in a closed flask. The resultant product (black) was completely rinsed with ethanol and dried for 24 h under vacuum at 35 °C [23].

### 2.4. Functionalization of MWCNTs

The treatment of pristine MWCNTs was carried out to obtain –COOH groups on the surfaces of the nanotubes. First, 50 mg of raw MWCNTs was heated in an acidic mixture containing 30 mL of conc. HNO_3_ and 90 mL of conc. H_2_SO_4_ for 5 h at 55 °C and then 8 h at 80 °C. This mixture was washed several times with distilled water until a neutral pH was achieved. The obtained solid particles were dried at 70 °C for 24 h [24].

### 2.5. Fabrication of NiPcMWCNTs Nanocomposites

NiPcMWCNTs nanocomposites were prepared (as illustrated in Figure 1) by dissolving 5.0 mg of NiNp and 4.0 mg of 29H, 31H phthalocyanine Pc in 1 mL DMF under ultrasonication for 3 h. Afterward, 4.0 mg of functionalized Multiwalled carbon nanotubes fMWCNTs was added to the DMF mixture and sonicated for an additional 5 h. The nickel phthalocyanine nanoparticles (NiPc) were then adsorbed onto fMWCNTs by spontaneous adsorption and sonication to obtain NiPcMWCNTs nanocomposites [25].

### 2.6. Electrode Modification Procedure

Before the modification, the glassy carbon electrode (GCE) surface was cleaned by gently polishing it with an aqueous slurry of alumina nanopowder on a micro cotton pad. Afterward, the cleaned GCE was thoroughly rinsed in distilled water and 100% ethanol for 5 min under ultrasonic vibration to eliminate any remaining remnants of alumina particles on the GCE’s surface. Nanocatalyst suspensions were produced by dissolving 3 mg of each of the prepared NiNp, Pc, fMWCNTs, and NiPcMWCNTs in 1 mL of DMF and subjecting the mixtures to 20 min of sonication. The bare GCE was modified with the prepared nanocatalyst suspensions by a drop-dry method. Five milliliters of the suspension of Ni, Pc, fMWCNTs, and NiPcMWCNTs each was dropped on the cleaned GCE and dried for 5 min at 50 °C.

### 2.7. Characterization of Fabricated Nanomaterials

The successful synthesis of the nanomaterials and the nanocomposite was confirmed by X-ray diffraction spectrophotometry, UV–visible spectrophotometry, scanning electron microscopy, and transmission electron microscopy techniques. Likewise, cyclic voltammetry (CV) and electrochemical impedance spectroscopy (EIS) were used to confirm the successful modification of the electrodes.

### 2.8. Electrochemical and Impedance Studies

Electrochemical characterization of the modified electrodes was performed by CV using 5 mM [Fe(CN)_6_]^4−/3−^ in 0.1 M PBS (pH 7). Impedance spectroscopy studies were performed in four different bromate electrolyte solutions: 0.1 M KBrO_3_ prepared in 0.1 M PBS (pH 7), 0.1 M KBrO_3_ prepared in 0.1 M sodium acetate buffer (SAB) (pH 7), 0.1 M KBrO_3_ prepared in 0.1 M Na_2_SO_4_ (pH 2), and 0.1 M KBrO_3_ prepared in 0.1 M H_2_SO_4_ (pH 1) at a constant potential (+0.2 V) and a frequency range between 100 kHz and 0.1 Hz. The bare and the modified electrodes (GCE-Ni, GCE-Pc, GCE-fMWCNTs, and GCE-NiPcMWCNTs) served as the working electrodes. The reference electrode used was Ag/AgCl in saturated KCl, while the counter electrode was platinum wire. All experiments were carried out at 25 ± 1 °C.

## 3. Results and Discussion

### 3.1. X-ray Diffraction XRD Characterization

XRD, a reliable technique, was employed to determine the crystalline structure of the prepared nickel nanoparticles (NiNp). The XRD pattern of NiNp, as illustrated in Figure 2a, shows three distinct peaks at 2θ = 44.5°, 51.9°, and 76.5°, corresponding to the (1 1 1), (2 0 0), and (2 2 0) planes of pure face-centered cubic nickel. These peaks are consistent with those described in other reports [23,26,27], suggesting that the synthesized nanoparticles are pure nickel nanoparticles. An average crystallite size (D) of 12 nm was obtained for the synthesized NiNp using Scherrer’s equation [28].
(1)$$D=\frac{0.89\lambda }{B\;\cos\;\theta }$$
where λ denotes the wavelength (0.15418 nm), B represents the full width at half-maximum peak, and θ stands for the XRD peak’s Bragg angle. 

Figure 2b shows the XRD spectra of the raw MWCNTs and functionalized MWCNTs. The diffraction peak at 2θ of 26.34° (002) is assigned to rMWCNTs, while 25.98° (002) and 42.32° (101) are for fMWCNTs. A well-pronounced diffraction peak observed at 2θ of 25.98° indicates the successful functionalization of the rMWCNTs [29,30]. Figure 2c illustrates the XRD patterns of NiPcMWCNTs, with eight peaks at 6.91°, 10.98°, 15.07°, 17.07°, 26.25°, 44.65°, 52.03° and 76.61°. The wide and strong peak at 26.25° (002) is attributed to the fMWCNTs, while 44.65°, 52.03° and 76.61° are assigned to nickel nanoparticles. The peaks at 6.91°, 10.98°, 15.07°, and 17.07° are characteristic peaks of phthalocyanines [29,30].

### 3.2. Energy Dispersive X-ray EDX Analysis

The EDX spectra depicted in Figure 3 portray the elemental composition of (a) nickel nanoparticles, (b) fMWCNTs, and (c) NiPcMWCNTs nanocomposites. The carbon and oxygen in the NiNp spectrum (Figure 3a) are impurities that arose from the precursor. Likewise, the appearance of sulfur in (Figure 3b) resulted from the sulfuric acid used during MWCNTs functionalization. The successful fabrication of the NiPcMWCNTs is confirmed by the presence of carbon (indicating MWCNTs), nitrogen (indicating phthalocyanine), and nickel in the nanocomposite’s spectrum (Figure 3c).

### 3.3. Ultraviolet–Visible Characterization

Figure 4 illustrates the UV–vis absorption spectra of (a) NiNp, (b) Pc, (c) rMWCNTs and fMWCNTs, and (d) NiPcMWCNTs nanocomposite. Figure 4a shows the absorption spectrum of NiNp with a characteristic absorption peak at 296 nm. Figure 4b reveals a sharp difference in the absorption peaks of rMWCNTs and fMWCNTs at 289 and 294 nm, confirming the successful functionalization of raw MWCNTs. The interaction between the overlapping orbitals of the central metal atom [31] and molecular orbitals of the 18 electrons of the aromatic system leads to the phthalocyanine UV-vis spectrum [25].

This spectrum is generated by electronic transitions between the highest occupied molecular orbital (HOMO) and the lowest unoccupied molecular orbital (LUMO), i.e., π* → π or π → π* transitions. Phthalocyanine exhibited two intense absorption bands, the B-band and Q-band. The B-band occurs in the UV region between 335–450 nm and results from π* → π transitions, while the Q-band occurs at 667 nm due to π → π* transitions. The D_2h_ symmetry of the compound led to the Q-band splitting in the spectrum, which occurred at 667 nm. The absorption spectra of the NiPcMWCNTs nanocomposite, as shown in Figure 4d, exhibit similar splitting of Q-bands (but broad) with all the observed peaks for NiNp, fMWCNTs, and Pc.

### 3.4. Transmission Electron Microscopy TEM Characterization

The size and morphology of the fabricated nanomaterials were characterized by TEM. TEM images of the NiNp, Pc, fMWCNTs, and NiPcMWCNTs nanocomposite are displayed in Figure 5a–d, respectively. In Figure 5a, the TEM image of NiNp shows a non-spherical particle shape with smooth but irregular particle morphology, having an average crystallite size of 18 nm, which is closer to the value obtained using Scherrer’s equation. The TEM image of the NiPcMWCNTs nanocomposite illustrated in Figure 5d shows larger aggregates of Pc and NiNp unevenly dispersed on the fMWCNTs’ surfaces.

### 3.5. Scanning Electron Microscopy SEM Characterization

Figure 6a–d portray the morphology and microstructures of SEM images of NiNp, Pc, fMWCNTs, and NiPcMWCNTs nanocomposite, respectively. The micrograph shows the attachment of both NiNp and Pc on the surface of the fMWCNTs and subsequently the successful formation of NiPcMWCNTs nanocomposite.

### 3.6. Electrochemical Characterization of Bare and Modified Electrodes

Cyclic voltammetry is a valuable technique for determining the electron transport properties of electrochemical materials. The characterization of the modified electrodes was performed using CV. As shown by the cyclic voltammograms (Figure 7), both the bare and modified electrodes displayed a pair of redox peaks in 5 mM [Fe(CN)_6_]^3−/4−^ prepared in 0.1 M PBS at pH 7 (scan rate of 25 mV s^−1^). Individual peak separation (Ep) values of 0.39, 0.25, 0.18, 0.13, and 0.18 V were obtained for the GCE, GCE-Pc, GCE-Ni, GCE-fMWCNTs, and GCE-NiPcMWCNTs, respectively, as shown in Table 1. These values are higher than the anticipated 0.059 V for a fast one-electron transfer, confirming the reaction at the electrodes as quasi-reversible [32,33]. Table 1 further shows that all electrodes have I_pa_/I_pc_ values of approximately one. This suggests that all reactions on the GCE and modified electrodes’ surfaces were reversible. The voltammogram in Figure 7 illustrates the comparative current response of the GCE with the modified electrodes in decreasing order of GCE-NiPcMWCNTs (457.0 μA) > GCE-fMWCNTs (58.0 μA) > GCE-Ni (35.6 μA) > GCE-Pc (21.5 μA) > GCE (25.0 μA). A higher current response of anodic peak current (I_pa_) of 457.0 μA was observed for the GCE-NiPcMWCNTs-modified electrode, which is about 18 times greater than those obtained with the GCE and other modified electrodes. This implies that the GCE-NiPcMWCNTs electrode has the fastest electron transport of all the electrodes. The high current response and enhanced electron transport of the GCE-NiPcMWCNTs-modified electrodes are attributed to the combined cooperative effect between NiNp, Pc, and fMWCNTs, which is the result of the larger surface area of Pc and fMWCNTs, as well as the strong electrical conductivity of the fMWCNTs and nickel. [25].

NiPcMWCNTs-modified electrodes exhibited a huge capacitive current, suggesting their high charge/discharge ability compared to the other electrodes. The specific capacitance (Cp) of the modified electrodes was calculated from the cyclic voltammograms using the equation:(2)$$\mathrm{Cp}=\frac{A}{2\mathrm{mk}\left(V_{2}\;–V_{1}\right)}$$
where A represents the area inside the CV curve in A V, k represents scan rate in mV/s, m represents the mass of active material in g, and (V_2_ − V_1_) represents the potential window in V/s. 

The Cp values, as illustrated in Table 1, were found to be 6.80, 0.63, 0.31, 0.26, and 0.17 F g^−1^ for GCE-NiPcMWCNTs, GCE-fMWCNTs, GCE-Ni, GCE-Pc, and GCE, respectively. From this result, it is obvious that GCE-NiPcMWCNTs electrodes produced a much higher specific capacitance value than the other modified electrodes. The improved capacitive behavior exhibited by the GCE-NiPcMWCNTs-modified electrode was also credited to the greater surface area of the fMWCNTs and phthalocyanine, as well as the high electrical conductivity of nickel and the fMWCNTs. The capacitive properties of these electrodes were later examined via the EIS technique.

#### 3.6.1. Scan Rate Study

CV was utilized to investigate the effect of scan rate (25–300 mVs^−1^) on the electrochemical reactions of the GCE-NiPcMWCNTs’ surface in 5 mM [Fe(CN)_6_]^3−/4−^ solution prepared in 0.1 M PBS at pH 7. Figure 8a shows that an increase in peak currents is directly proportional to the scan rate. A linear variation exists in the plot of the peak currents versus the square root of the scan rate (Figure 8b), yielding the following regression equation:I_pa_ = 0.0054 v^1/2^ − 0.0004; R^2^ = 0.9987
I_pc_ = −0.0048 v^1/2^ + 0.0003; R^2^ = 0.999

GCE-NiPcMWCNTs exhibit a typical diffusion-controlled mechanism, as evidenced by the linearity between the square root of the scan rate and peak currents.

#### 3.6.2. Stability Study

With 20 repeated scans (25 mVs^−1^) in 5 mM [Fe(CN)_6_]^3−/4−^ solution produced in 0.1 M PBS at pH 7, the stability of GCE-NiPcMWCNTs was investigated. A slight current drop (4.3%) was exhibited by the GCE-NiPcMWCNTs-modified electrode, as shown in Figure 9, suggesting the stability of the electrode.

### 3.7. Impedance Spectroscopy Studies of Bare and Modified Electrodes

To extensively study both the electron transport and capacitive properties of the GCE and modified electrodes, an EIS experiment was conducted in four different electrolytes using an Autolab Potentiostat PGSTAT 302 at a frequency range of 100 kHz–0.1 Hz and a fixed potential of 0.2 V versus Ag/AgCl, saturated with 3 M KCl. The highest capacitance was found at a voltage of 0.2 V; thus, that was used for this experiment. The various electrolytes used for this study include: 0.1 M KBrO_3_ solution in 0.1 M PBS (pH 7), 0.1 M KBrO_3_ solution in 0.1 M sodium acetate buffer SAB (pH 7), 0.1 M KBrO_3_ solution in 0.1 M Na_2_SO_4_ (pH 2), and 0.1 M KBrO_3_ solution in 0.1 M H_2_SO_4_ (pH 1). The Metrohm Autolab NOVA 2.1.3 software (Utrecht, The Netherlands) was used to fit all impedance data.

#### 3.7.1. EIS of Modified Electrodes in 0.1 M KBrO_3_ Prepared in 0.1 M PBS

Figure 10 shows the fitted Nyquist plots of the modified and unmodified GCE electrodes in 0.1 M KBrO_3_ solution produced in 0.1 M PBS (pH 7) electrolytes. Figure 10b,c represent the circuit diagram of the EIS data fitting. In the circuit, Rs denotes solution resistance, R_ct_ denotes charge-transfer resistance, C denotes capacitance, CPE denotes constant phase element, and W denotes Warburg impedance. The summary of the fitted impedance data with their chi-square x^2^ values is shown in Table 2. The negative x^2^ values and the low percentage errors (in brackets) indicate that the EIS data were successfully fitted. The fitted impedance data in Table 2 reveal that the GCE-NiPcMWCNTs electrode has the lowest R_ct_ value of 1.98 kΩ, suggesting a fast electron transfer of the GCE-NiPcMWCNTs electrode compared to the other electrodes. The high conductivity of the functionalized MWCNTs, acting as a good electron conductor between NiNp and Pc and the electrode surface, accounts for the GCE-NiPcMWCNTs’ fast electron transport. According to the results, electron transport was significantly faster at the GCE-NiPcMWCNTs electrode, aligning with CV results. This also contributed to the high Cp value of the GCE-NiPcMWCNTs electrode. The n values ranging from 0.6–0.8 for all electrodes may be attributable to the ease with which ions diffuse to and from the electrode/solution contact.

Further investigation on the capacitive behaviors of the electrodes was carried out using the “knee” frequency (f°). The frequency at which capacitive behavior is prominent is referred to as the “knee” frequency. It is a measure of a supercapacitor’s power capability, as it is the transition point between the high-frequency and low-frequency components. The greater the f°, the faster the supercapacitor can charge and discharge, or the higher the power density the supercapacitor can attain [34,35]. The f° value for the GCE-NiPcMWCNTs in PBS (2.22 Hz), as depicted in Table 2, is approximately two times higher than that of the other electrodes. The figure is fairly significant, and it validates the electrode’s high Cp values in the electrolytes. Most commercially available supercapacitors, particularly those designed for greater power applications, work at rates less than 1 Hz, according to a study [3].

#### 3.7.2. EIS of Modified Electrodes in 0.1 M KBrO_3_ Prepared in 0.1 M H_2_SO_4_

Figure 11a depicts the fitted Nyquist plots of the modified and unmodified GCE electrodes in 0.1 M KBrO_3_ prepared in 0.1 M H_2_SO_4_ (pH 1) electrolyte, with Figure 11b representing the circuit diagram used for the EIS data fitting. In Table 2, the R_ct_ values obtained for the electrodes in H_2_SO_4_ electrolyte are in increasing order of 0.06 < 2.4 < 6.6 < 7.7 < 13.4 KΩ for GCE-NiPcMWCNTs, GCE-fMWCNTs, GCE-Pc, GCE-Ni, and GCE, respectively, with GCE-NiPcMWCNTs exhibiting the lowest R_ct_ value, suggesting its fastest electron transport ability compared to the other electrodes in the electrolyte. From this result, the GCE-NiPcMWCNTs electrode exhibited faster electron transport in H_2_SO_4_ than other electrodes_,_ similar to what was obtained for PBS. f° values of 5.18, 1.27, 2.95, 9.10, and 21.2 Hz were obtained in H_2_SO_4_ electrolyte for the GCE, GCE-Pc, GCE-Ni, GCE-fMWCNTs, and GCE-NiPcMWCNTs, respectively, with the GCE-NiPcMWCNTs electrode having the highest f° value (21.2 Hz), thus suggesting a higher power capability of the electrode.

#### 3.7.3. EIS of Modified Electrodes in 0.1 M KBrO_3_ Prepared in 0.1 M Na_2_SO_4_

Figure 12 presents the fitted Nyquist plots for the GCE and modified electrodes generated in 0.1 M KBrO_3_ prepared in 0.1 Na_2_SO_4_ (pH 2) electrolyte, with Figure 12b indicating the circuit diagram used for the EIS data fitting. The trend of the EIS results obtained for the electrodes in Na_2_SO_4_ electrolyte is not very different from the two previous electrolytes, with the GCE-NiPcMWCNTs electrode exhibiting the fastest electron transport ability. The R_ct_ values obtained in Na_2_SO_4_ electrolyte were 0.61, 0.69, 7.1, 10.1, and 17.3 kΩ for GCE-NiPcMWCNTs, GCE-fMWCNTs, GCE-Ni, GCE-Pc, and GCE, respectively, with GCE-NiPcMWCNTs demonstrating their usual faster electron transport compared to the other electrodes in the electrolyte. An f° value of 6.87 Hz was obtained for the GCE-NiPcMWCNTs electrode in Na_2_SO_4_ electrolyte. This value is higher than the values for other electrodes, as expected, which is indicative of its higher power capability.

#### 3.7.4. EIS of Modified Electrodes in 0.1 M KBrO_3_ Prepared in 0.1 M SAB (pH 7)

Figure 13 presents the fitted Nyquist plots for the GCE and modified electrodes generated in 0.1 M KBrO_3_ prepared in 0.1 SAB (pH 7) electrolyte, with Figure 13b indicating the circuit diagram used for the EIS data fitting.

Expectedly, the GCE-NiPcMWCNTs electrode exhibited the fastest electron transport behavior in SAB electrolyte with a lower R_ct_ value of 0.36 kΩ. Additionally, a similar trend was observed in their f° values. The f° values obtained in SAB were in increasing order of GCE (0.54) = GCE-fMWCNTs (0.54) < GCE-Pc (0.95) < GCE-Ni (1.27) < GCE-NiPcMWCNTs (1.68). The GCE-NiPcMWCNTs electrode demonstrated a greater f° value (1.68 Hz) compared to the other modified electrodes, which is nearly three times greater than that of the unmodified GCE.

#### 3.7.5. Comparison of EIS Results of GCE-NiPcMWCNTs-Modified Electrode in Different Electrolytes

A comparative summary (Figure 14) of the EIS results of the GCE-NiPcMWCNTs electrode in four different electrolytes is given in Table 3. As shown in Table 3, R_ct_ values of 1.98, 0.06, 0.61, and 0.36 kΩ were obtained for the GCE-NiPcMWCNTs electrode in PBS, H_2_SO_4,_ Na_2_SO_4,_ and SAB electrolyte, respectively. GCE-NiPcMWCNTs had the lowest R_ct_ value (0.06 kΩ) in H_2_SO_4_ electrolyte compared with the other electrolytes. This shows that the fastest electron transport capability of GCE-NiPcMWCNTs was achieved in H_2_SO_4_ electrolyte, and the slowest was obtained in PBS. 

The f° values obtained for the GCE-NiPcMWCNTs in PBS, H_2_SO_4_, Na_2_SO_4_, and SAB electrolyte were 2.22, 21.2, 6.87, and 1.68 Hz, respectively. The f° value for H_2_SO_4_ (21.2 Hz) is about three times higher than that for Na_2_SO_4_ (6.87 Hz). SAB had the lowest f° value (Table 3). The figure is highly important and further validates the high electron transport capability of GCE-NiPcMWCNTs in all the electrolytes. The higher electron transport properties and higher f° values of GCE-NiPcMWCNTs electrode in H_2_SO_4_ electrolytes were attributed to fewer H^+^ ions in the acid.

Phase angles of ~74.0°, 42.0°, 36.0°, and 33.3° were obtained for the GCE-NiPcMWCNTs electrode in SAB, H_2_SO_4,_ PBS, and Na_2_SO_4_ electrolytes, respectively, from their individual Bode plots of phase angle (°) vs. log f° at −1.0 Hz, as illustrated in Figure 15. A lower phase angle (33.3°) was observed at the same frequency in Na_2_SO_4_ electrolyte compared to the other electrolytes. The values of the phase angles of GCE-NiPcMWCNTs in all the electrolytes were less than the anticipated 90° for an ideal capacitive property. This is a further confirmation of the GCE-NiPcMWCNTs as a non-ideal capacitor.

Similarly, slope values of −0.2, −0.3, −3.2, and −5.3 were obtained from the graphs of log |Z| vs. log f° for H_2_SO_4_, Na_2_SO_4_, PBS, and SAB electrolytes, respectively. These values are less than the expected slope of 1.0 for pure capacitive behaviors, affirming that the GCE-NiPcMWCNTs electrode investigated in this research is a high-specific-capacitance, high-energy-density hybrid supercapacitor that combines the faradaic pseudocapacitive and electric double-layer system.

## 4. Conclusions

This study demonstrated the successful fabrication and characterization of a hybrid supercapacitor, as confirmed by UV/vis spectroscopy, XRD, EDX, SEM, and TEM techniques. CV and EIS studies showed that the GCE-NiPcMWCNTs-modified electrode exhibited rapid electron transfer ability and higher supercapacitive behavior in H_2_SO_4_ than the other electrodes. A similar trend in power density was observed for the GCE-NiPcMWCNTs, which had the highest power density, decreasing in the order of H_2_SO_4_ > Na_2_SO_4_ > PBS > SAB. The hybrid supercapacitor also exhibited high electrochemical stability with insignificant changes (4.3%) over 20 repeated scan rates. Hence, GCE-NiPcMWCNTs-modified electrode exhibited high energy and power densities and high specific capacitance. The enhanced electron transport capability and capacitive properties of the GCE-NiPcMWCNTs in H_2_SO_4_ compared to the other electrolytes were credited to relatively fewer H^+^ ions in the acid, allowing for easier ion penetration in the acidic electrolyte. The improved capacitive behavior of NiPcMWCNTs was also credited to the greater surface area of the fMWCNTs and phthalocyanine and the high electrical conductivity of the nickel and fMWCNTs. Finally, the findings of this study show that GCE-NiPcMWCNTs is a suitable electrochemical sensor for detecting bromate, especially in H_2_SO_4_ and Na_2_SO_4_ electrolyte solutions.

## Figures and Tables

**Figure 1 nanomaterials-12-01876-f001:**
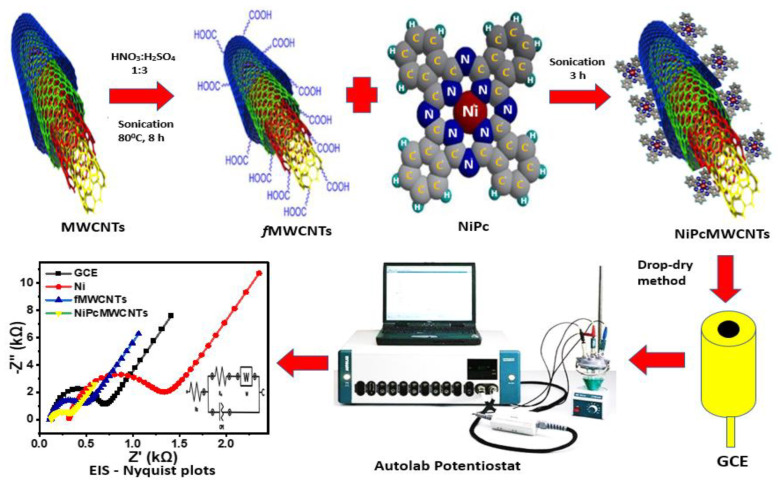
Schematic diagram for the fabrication, modification, and EIS characterization of GCE-NiPcMWCNTs-modified electrode.

**Figure 2 nanomaterials-12-01876-f002:**
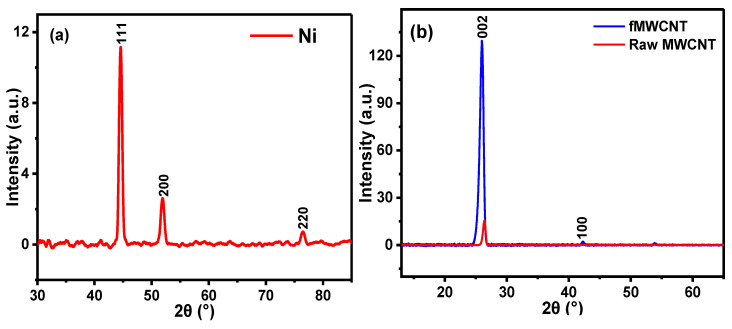
XRD spectra of (**a**) NiNp, (**b**) rMWCNTs and fMWCNTs, and (**c**) NiPcMWCNTs nanocomposite.

**Figure 3 nanomaterials-12-01876-f003:**
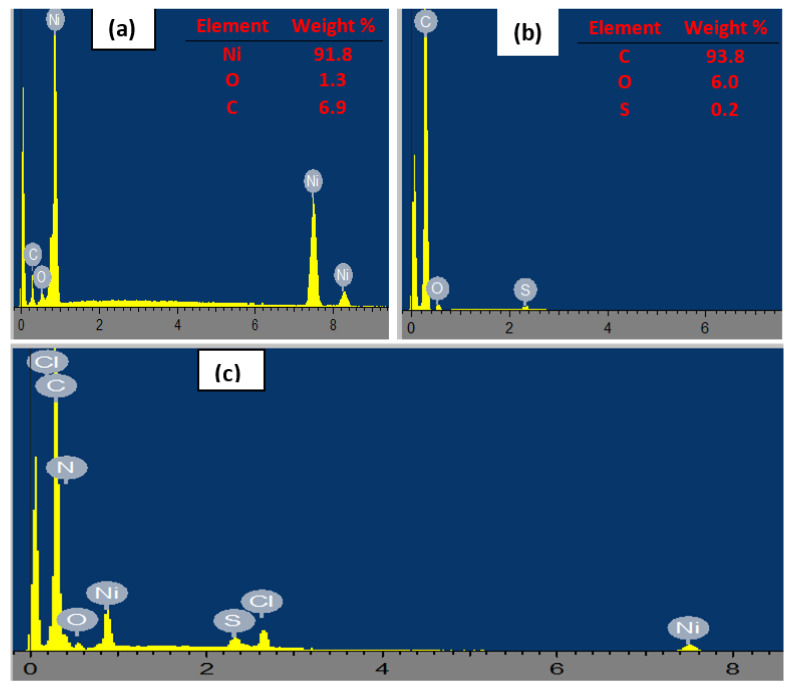
EDX spectra of (**a**) NiNp, (**b**) fMWCNTs, and (**c**) NiPcMWCNTs nanocomposite.

**Figure 4 nanomaterials-12-01876-f004:**
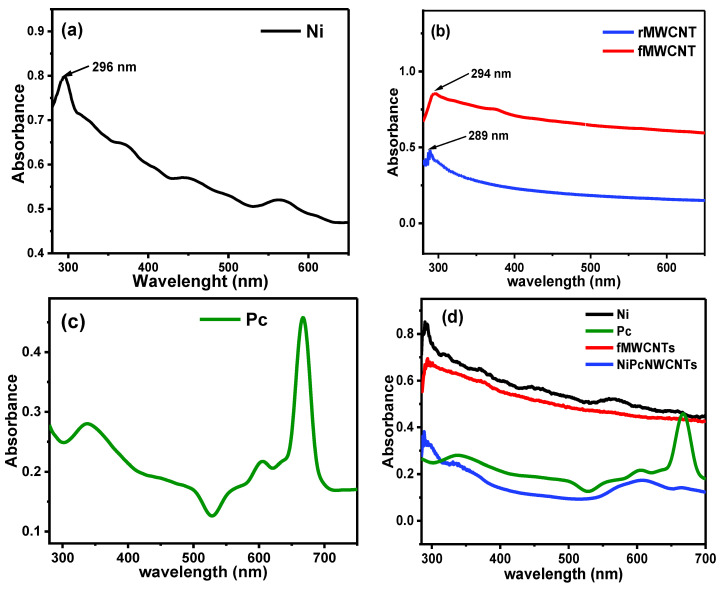
UV–vis spectra for (**a**) NiNp, (**b**) rMWCNTs and fMWCNTs, (**c**) Pc, and (**d**) comparative spectrum for NiNp, Pc, fMWCNTs, and NiPcMWCNTs nanocomposite.

**Figure 5 nanomaterials-12-01876-f005:**
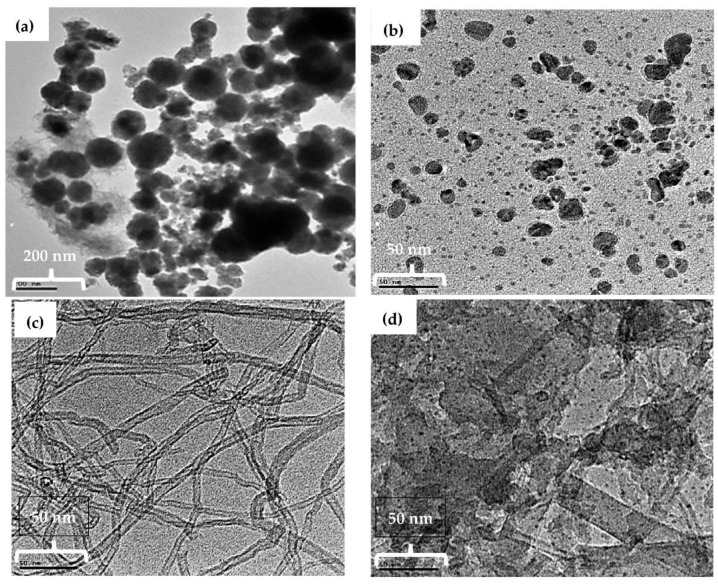
TEM images of (**a**) NiNp, (**b**) Pc, (**c**) fMWCNTs, and (**d**) NiPcMWCNTs nanocomposite with the size of scale bar: Ni (200 nm), Pc (50 nm), fMWCNTs (50 nm), and NiPcMWCNTs (50 nm).

**Figure 6 nanomaterials-12-01876-f006:**
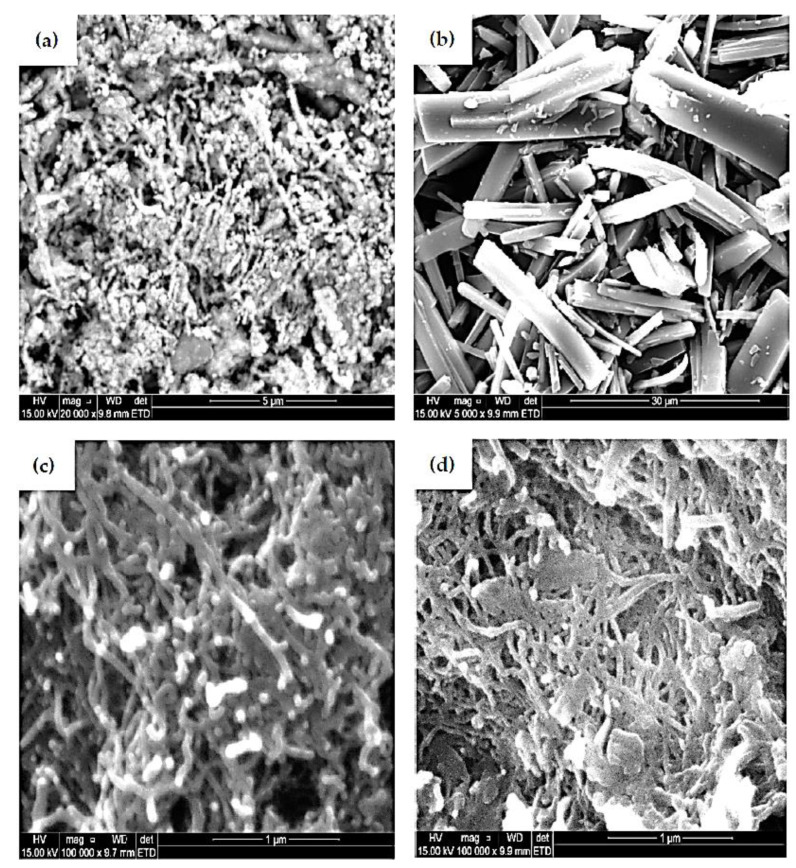
SEM images of (**a**) NiNp, (**b**) Pc, (**c**) fMWCNTs, and (**d**) NiPcMWCNTs nanocomposite.

**Figure 7 nanomaterials-12-01876-f007:**
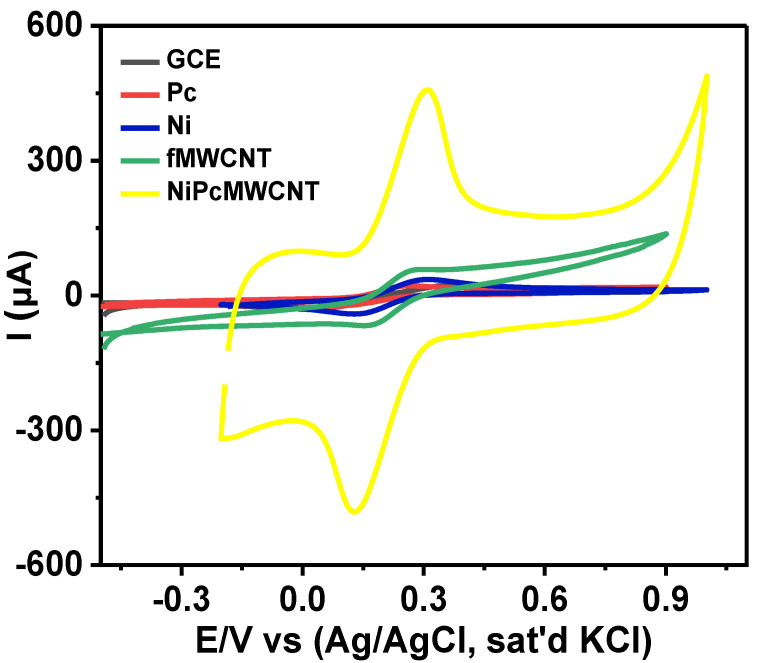
Comparative cyclic voltammograms of bare and modified electrodes in 5 mM Fe(CN)_6_]^3−/4−^ prepared in 0.1 M PBS at pH 7 (scan rate 25 mVs^−1^).

**Figure 8 nanomaterials-12-01876-f008:**
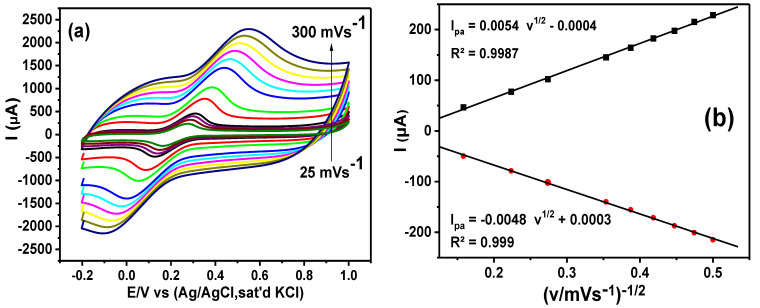
(**a**) Cyclic voltammograms of GCE-NiPcMWCNTs (scan rate, 25–300 mVs^−1^) and (**b**) the linear graph of the peak current versus square root of scan rate in 5 mM [Fe(CN)_6_]^3−/4−^ solution prepared in 0.1 M PBS.

**Figure 9 nanomaterials-12-01876-f009:**
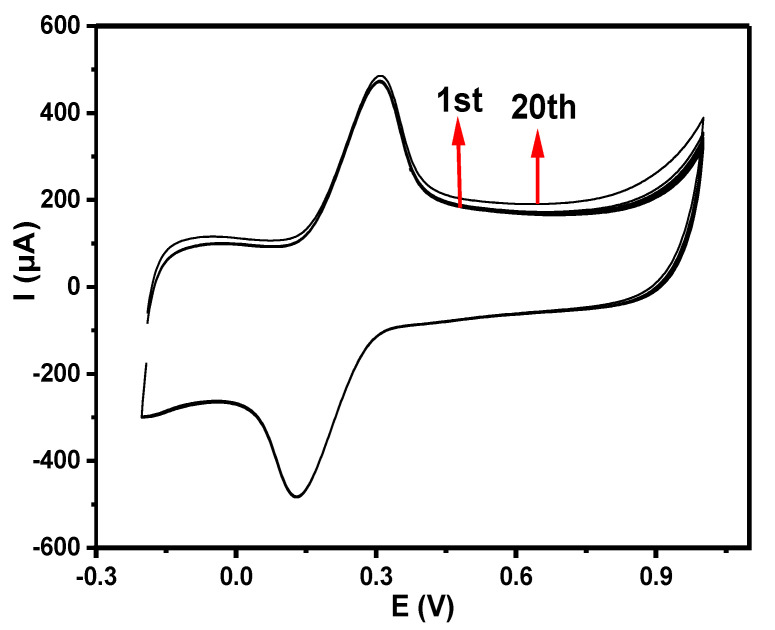
Cyclic voltammograms of 20 scans of GCE-NiPcMWCNTs-modified electrode in 5 mM [Fe(CN)_6_]^3−/4−^ solution produced in 0.1 M PBS at pH 7 (scan rate 25 mVs^−1^).

**Figure 10 nanomaterials-12-01876-f010:**
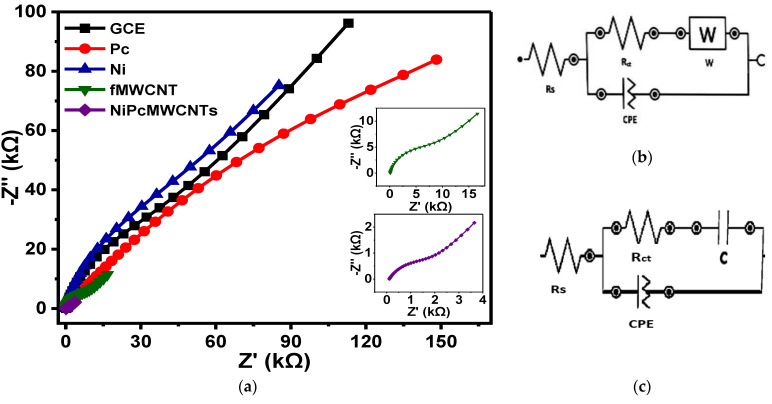
(**a**) Nyquist plots of GCE and modified electrodes in 0.1 M KBrO_3_ solution produced in 0.1 M PBS (pH 7.0). (**b**,**c**) The circuit used for fitting the EIS data (**b**) for GCE, Ni, fMWCNTs, and NiPcMWCNTs and (**c**) for Pc.

**Figure 11 nanomaterials-12-01876-f011:**
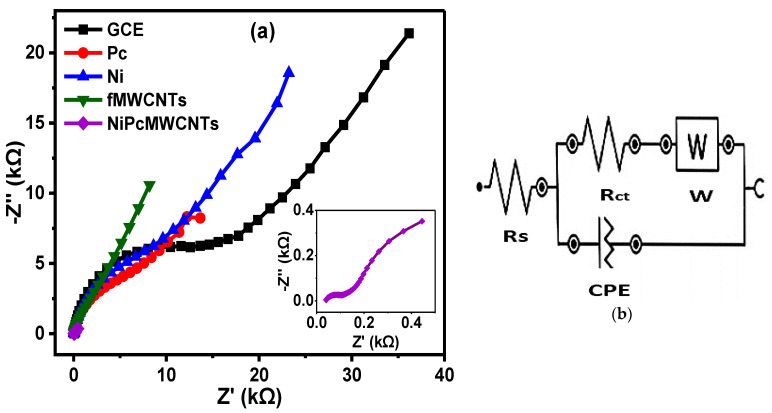
(**a**) Nyquist plots of GCE and modified electrodes in 0.1 M KBrO_3_ solution produced in 0.1 M H_2_SO_4_ (pH 1) and (**b**) is the circuit used for EIS data fitting.

**Figure 12 nanomaterials-12-01876-f012:**
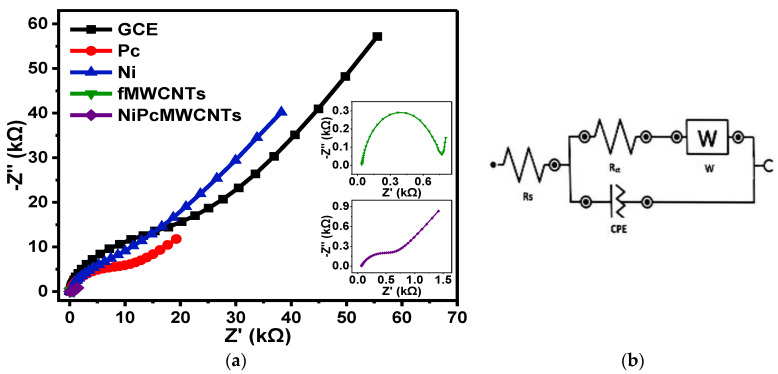
(**a**) Nyquist plots of GCE and modified electrodes in 0.1 M KBrO_3_ produced in 0.1 M Na_2_SO_4_ (pH 2). (**b**) represents the circuit used for the EIS data fitting.

**Figure 13 nanomaterials-12-01876-f013:**
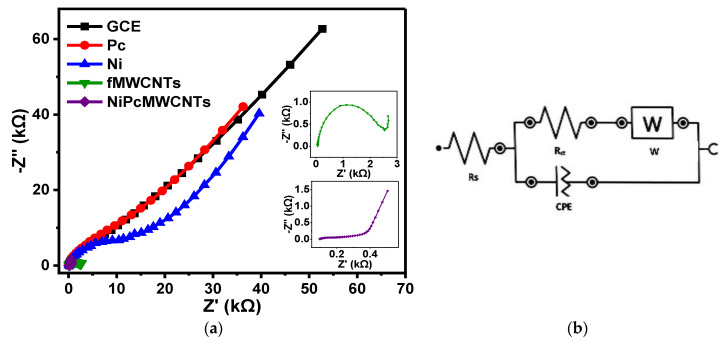
(**a**) Nyquist plots of modified and unmodified GCE in 0.1 M KBrO_3_ solution produced in 0.1 M SAB (pH 7). (**b**) The circuit used for EIS data fitting.

**Figure 14 nanomaterials-12-01876-f014:**
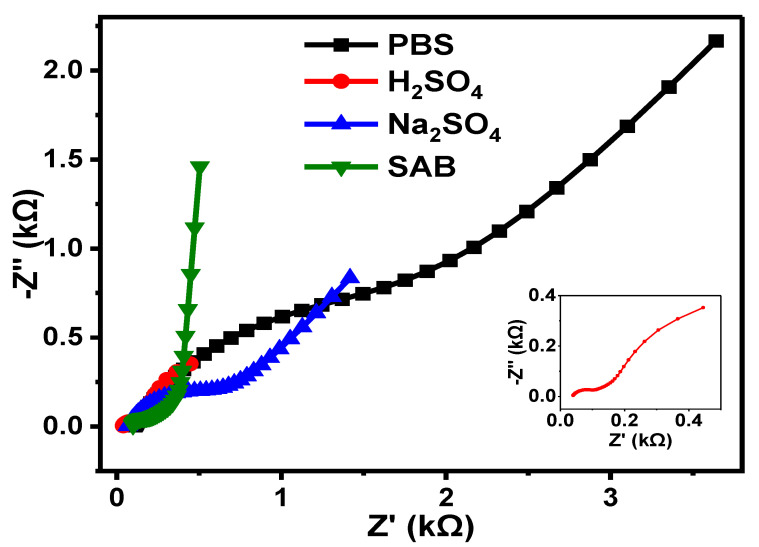
Nyquist plots of GCE-NiPcMWCNTs electrode in PBS, H_2_SO_4_, Na_2_SO_4_, and SAB electrolytes.

**Figure 15 nanomaterials-12-01876-f015:**
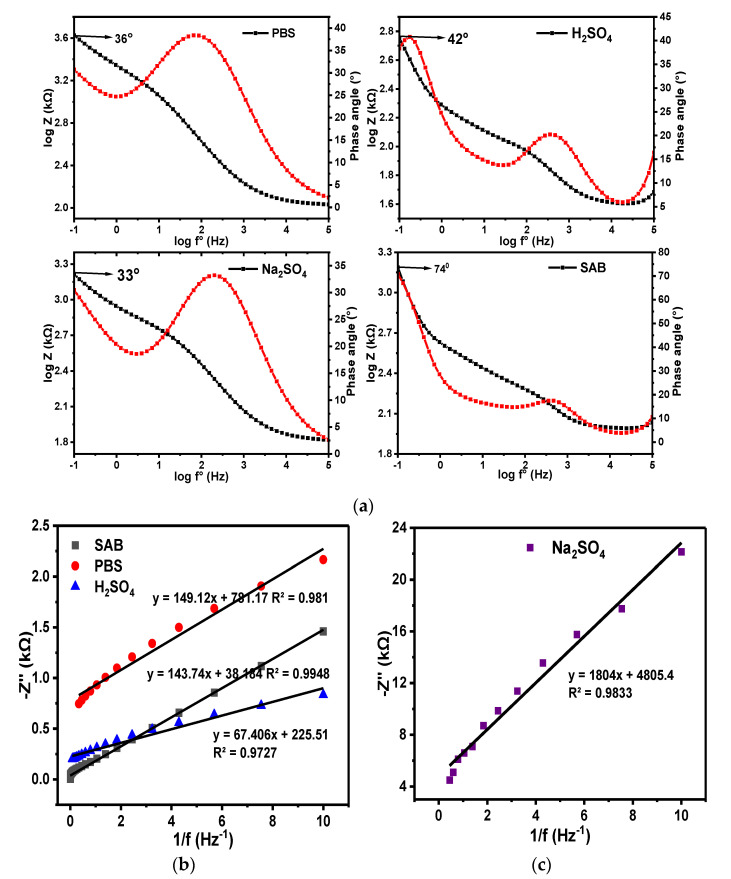
(**a**) Bode plots of −phase angle (°) vs. log f° and plots of log Z vs. log f° for GCE-NiPcMWCNTs in PBS, H_2_SO_4_, Na_2_SO_4_, and SAB electrolytes. (**b**,**c**) are the corresponding plots of −Z″ vs. 1/ f°.

**Table 1 nanomaterials-12-01876-t001:** Summary of CV Results of Bare and Modified Electrodes in 5 mM [Fe(CN)_6_]^3−/4−^ Prepared in 0.1 M PBS at pH 7 (Scan Rate, 25 mVs^−1^).

Working Electrodes	I_pa_ (μA)	I_pc_ (μA)	I_pa_/I_pc_	E_pa_ (V)	E_pc_ (V)	ΔEp (V)	E° (V)	Cp (F/g)
Bare GCE	25.0	−30.0	−0.83	0.40	0.01	0.39	0.20	0.17
GCE-Pc	21.5	−25.9	−0.83	0.26	0.01	0.25	0.13	0.26
GCE-Ni	35.6	−41.3	−0.86	0.31	0.13	0.18	0.09	0.31
GCE-fMWCNTs	58.0	−67.1	−0.87	0.29	0.16	0.13	0.07	0.63
GCE-NiPcMWCNTs	457.0	−482.0	−0.95	0.31	0.13	0.18	0.09	6.80

**Table 2 nanomaterials-12-01876-t002:** Summary of fitted EIS results of bare and modified electrodes in different electrolytes.

Impedance Spectroscopy Data								
Electrolyte	Electrode	R_s_ (Ω)	CPE (μF)	R_ct_ (kΩ)	W (μF)	*N*	X^2^	f°
PBS	GCE	100 (2.73)	1.65 (6.09)	44.9 (9.92)	9.24 (5.44)	0.82 (0.95)	0.229	0.95
	Pc	174 (10.0)	4.85 (3.06)	428.2 (10.0)	37.6 (111)	0.49 (0.88)	0.137	0.72
	Ni	102 (1.60)	3.72 (3.96)	54.6 (9.50)	12.4 (6.51)	0.84 (0.70)	0.126	0.95
	fMWCNTs	93 (1.68)	8.08 (6.53)	7.95 (6.94)	82.6 (5.65)	0.89 (1.16)	0.200	1.27
	NiPcMWCNTs	105 (2.20)	43.4 (10.65)	1.98 (7.21)	407 (5.98)	0.63 (2.38)	0.164	2.22
H_2_SO_4_	GCE	35.3 (4.48)	0.55 (6.49)	13.4 (3.21)	36.7 (3.96)	0.87 (0.80)	0.216	5.18
	Pc	57.1 (1.56)	11.0 (5.33)	6.6 (5.89)	120 (5.06)	0.85 (0.94)	0.141	1.27
	Ni	40.3 (2.15)	3.75 (6.27)	7.7 (6.35)	47.6 (3.95)	0.88 (0.93)	0.188	2.95
	fMWCNTs	36.0 (2.08)	33.3 (13.4)	2.4 (38.09)	78.6 (5.48)	0.84 (2.29)	0.247	9.10
	NiPcMWCNTs	39.9 (2.95)	42.0 (53.2)	0.06 (9.53)	3130 (3.96)	0.79 (8.22)	0.320	21.2
Na_2_SO_4_	GCE	81.5 (2.50)	1.51 (7.70)	17.3 (8.65)	18.0 (4.57)	0.91 (1.10)	0.281	1.68
	Pc	87.8 (1.62)	5.87 (5.29)	10.1 (4.85)	80.2 (5.53)	0.85 (0.91)	0.149	1.27
	Ni	71.0 (1.59)	2.25 (6.71)	7.1 (8.11)	23.3 (2.36)	0.90 (0.94)	0.114	5.18
	fMWCNTs	40.2 (1.86)	16.2 (9.27)	0.69 (2.45)	8594 (23.9)	0.88 (1.51)	0.211	0.72
	NiPcMWCNTs	64.0 (5.29)	49.6 (9.2)	0.61 (10.8)	1060 (9.63)	0.65 (5.92)	0.071	6.87
SAB	GCE	132 (1.42)	1.55 (9.02)	6.35 (12.0)	15.7 (1.96)	0.92 (1.28)	0.102	0.54
	Pc	144 (1.13)	3.93 (6.26)	9.09 (10.1)	22.5 (2.39)	0.89 (0.99)	0.080	0.95
	Ni	120 (2.14)	0.85 (7.00)	11.1 (4.26)	24.8 (2.76)	0.87 (0.95)	0.150	1.27
	fMWCNTs	65 (1.71)	6.87 (7.11)	2.19 (2.47)	1465 (13.0)	0.91 (1.14)	0.188	0.54
	NiPcMWCNTs	91 (2.50)	306 (11.5)	0.36 (5.12)	714 (5.70)	0.50 (3.68)	0.192	1.68

**Table 3 nanomaterials-12-01876-t003:** Comparative summary of EIS results of GCE-NiPcMWCNTs-modified electrodes in all the electrolytes.

Impedance Spectroscopy Data							
Electrolytes	R_s_ (Ω)	CPE (μF)	R_ct_ (kΩ)	W (μF)	n	X^2^	f°
PBS	105 (2.20)	43.4 (10.65)	1.98 (7.21)	407 (5.98)	0.63 (2.38)	0.164	2.22
H_2_SO_4_	39.9 (2.95)	42.0 (53.2)	0.06 (9.53)	3130 (3.96)	0.79 (8.22)	0.320	21.2
Na_2_SO_4_	62.6 (1.96)	17.8 (8.06)	9.04 (21.8)	37.5 (5.88)	0.78 (1.49)	0.186	6.87
SAB	91 (2.50)	306 (11.5)	0.36 (5.12)	714 (5.70)	0.50 (3.68)	0.192	1.68

## Data Availability

Data are available upon request from authors.

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
