# Peer review of "Effects of Electrolytes on the Electrochemical Impedance Properties of NiPcMWCNTs-Modified Glassy Carbon Electrode"

_nanomaterials, 2022, doi:10.3390/nano12111876_

Round 1

Reviewer 1 Report

The manuscript ID: nanomaterials-1713962, entitled “Effects of Electrolytes on the Electrochemical Impedance Properties of NiPcMWCNTs Modified Glassy Carbon Electrode” was carefully reviewed, and provided comments are listed as below. The manuscript reports the electrochemical characteristics of nickel phthalocyanine multiwalled carbon nanotubes nanocomposite on glassy carbon electrode (NiPcMWCNTs-GCE) in different electrolyte conditions. The results are interesting and important, and there are good connections between the reported data and the discussion part. However, the manuscript needs further improvement and I recommend the publication of this manuscript after modifying based on the provided comments.

  1. Experimental Section, Page 3, lines 100-108. Please exactly identify the company where raw materials were purchased.
  2. Page 5, line 175, Eq. 1. Please indicate if the instrumental broadening subtracted from the full width half maximum. If yes, that equation needs to be corrected.
  3. The main advantage of MWCNTs is the high surface area that provides more redox areas with electrolytes. However, the SEM images (particularly Fig. 5d) show that somehow that large surface area provided by MWCNTs is destroyed and decreased! Which provides contradiction with the discussion and results part in electrochemical tests. Please correct that.
  4. Fig. 2b related to XRD is somehow strange by comparing the intensity of raw MWCNTs and functionalized MWCNTs. By comparing the intensity of (002) peak it seems authors provide huge thickness (it can be considered as a huge agglomeration) on the raw MWCNTs.
  5. The best strategy for getting high electrochemical performance in carbon nanotubes is to keep the high surface area and at the same time deposit an ultrathin layer of highly conformal pseudocapacitance materials. The blow paper that is related to providing a highly conformal coating of pseudocapacitance on the carbon fibers with achieving a highly conformal coating needs to be mentioned and cited in the introduction section.

1) B. Li, H. Lopez-Beltran, C. Siu, K. H. Skorenko, H. Zhou, W. E. Bernier, M. Stanley Whittingham, and W. E. Jones, Jr. Vaper Phase Polymerized PEDOT/Cellulose Paper Composite for Flexible Solid-State Supercapacitor, ACS Appl. Energy Mater. 2020, 3, 2, 1559–1568.

2) M.H. Gharahcheshmeh, K. K. Gleason, Device Fabrication Based on oxidative Chemical Vapor Deposition (oCVD) Synthesis of Conducting Polymers and Related Conjugated Organic Materials. Adv. Mater. Interfaces. 2019, 6, 1801564.

Author Response

The Editor

Nanomaterials

REVISION OF MANUSCRIPT SUBMITTED FOR PUBLICATION

Manuscript Title:  Effects of Electrolytes on the Electrochemical Impedance Properties of NiPcMWCNTs Modified Glassy Carbon Electrode

Manuscript ID: nanomaterials-1713962  

We appreciate the reports on our manuscript. As a result of this, we submit a response to the comments on the manuscript for further consideration.

The reviewers' efforts at improving this manuscript are well appreciated. We have carefully considered the comments. The responses to all reviewers' comments are highlighted in yellow in the manuscript. 

Reviewer 1

  1. Experimental Section, Page 3, lines 100-108. Please exactly identify the company where raw materials were purchased.

Response: The companies where the raw materials were purchased have been included and are highlighted in yellow, see page 3, lines 107 - 111.

  1. Page 5, line 175, Eq. 1. Please indicate if the instrumental broadening subtracted from the full width half maximum. If yes, that equation needs to be corrected.

Response: No

  1. The main advantage of MWCNTs is the high surface area that provides more redox areas with electrolytes. However, the SEM images (particularly Fig. 5d) show that somehow that large surface area provided by MWCNTs is destroyed and decreased! Which provides contradiction with the discussion and results part in electrochemical tests. Please correct that.

Response: The SEM image of NiPcMWCNTs in Fig. 5d (now figure 6d) has been replaced with another one with a high resolution to show that the large surface area of the MWCNTs were not destroyed

  1. Fig. 2b related to XRD is somehow strange by comparing the intensity of raw MWCNTs and functionalized MWCNTs. By comparing the intensity of (002) peak it seems authors provide huge thickness (it can be considered as a huge agglomeration) on the raw MWCNTs.

Response: Though the intensity of (002) peak might not be the best way of comparing the raw MWCNTs and functionalized MWCNTs but it has been one of the parameters used in literatures eg

  • Mphuthi, N. G., Adekunle, A. S., Fayemi, O. E., Olasunkanmi, L. O., & Ebenso, E. E. (2017). Phthalocyanine doped metal oxide nanoparticles on multiwalled carbon nanotubes platform for the detection of dopamine. Scientific reports7(1), 1-23.

  • Nie, P., Min, C., Song, H. J., Chen, X., Zhang, Z., & Zhao, K. (2015). Preparation and tribological properties of polyimide/carboxyl-functionalized multi-walled carbon nanotube nanocomposite films under seawater lubrication. Tribology Letters58(1), 1-12.

  • Li, Y., Gan, G., Huang, Y., Yu, X., Cheng, J., & Liu, C. (2019). Ag-NPs/MWCNT composite-modified silver-epoxy paste with improved thermal conductivity. RSC advances9(36), 20663-20669.

We however confirm the presence of nickel nanoparticle by supplying data for the EDX spectral has been included to indicate the presence of Nickel nanoparticles in the NiPcMWCNTs composites. See figure 3, page 7, line 192 – 199.

  1. The best strategy for getting high electrochemical performance in carbon nanotubes is to

keep the high surface area and at the same time deposit an ultrathin layer of highly conformal pseudocapacitance materials. The blow paper that is related to providing a highly conformal coating of pseudocapacitance on the carbon fibers with achieving a highly conformal coating needs to be mentioned and cited in the introduction section.

Response: The suggestion has been included in the introduction section with references cited. Corrections are highlighted in yellow, see line 87 - 89, page 2.

Reviewer 2 Report

This manuscript submitted by Fayemi et al. describes the fabrication and property of a NiPcMWCNTs-modified GC electrode. The author found an interesting feature of the electrode in H2SO4. However, I would like to point out two criticisms relating to NiPcMWCNTs-modified electrode structure. Once these issues have been resolved, this paper may be publishable in Nanomaterials.

First, one might suspect that the NiPcMWCNTs-modified electrode is Ni(0)-nanoparticle and free-base phthalocyanine and MWCNTs deposited GC, not containing Ni-phthalocyanine complex. Usually, the preparation of metal phthalocyanine (M(II)Pc) is carried out by mixing with free-base phthalocyanine (H2Pc) with metal (II) species, such as Ni(OAc)2. I am unsure whether the metal introduction to the free-base phthalocyanine (H2Pc) occurs with Ni (0) nanoparticles or not. The formation of the Ni-phthalocyanine complex may be confirmed in the UV-Vis spectrum. However, the provided UV-Vis spectrum was indistinguishable because the spectrum presented may not be the spectrum in the solution.

Second, if the electrode contains Ni (0)-nanoparticle, structural changes would occur in dilute aqueous sulfuric acid, which might participate in the characteristic feature of the electrode in dilute aqueous sulfuric acid. The stability experiment of the electrode in sulfuric acid (pH1) is recommended in the same manner in PBS (pH7).

Here are my remarks:

1) line75: p => π?

2) line 102: sulphuric => sulfuric

3) line 103: K[Fe(CN)6]3-/4- => [Fe(CN)6]3-/4-

4) line 105: sodium hydrogen => sodium dihydrogen

5) line 160: 101 => 100?

6) line 205: Figure 3. Is the vertical axis absorbance?

7) line 336: Figure 9. Is the vertical axis -Z”?

8) line 354: Figure 10, caption. 0.1 M H2SO4 (pH1) => 0.05 M or pH0.7?

9) line 356 and etc.: BrO3 => H3BO3 or NaBrO3?

10) line 439: NAB => SAB?

11) line 470: HAB => SAB?

Author Response

The Editor

Nanomaterials

REVISION OF MANUSCRIPT SUBMITTED FOR PUBLICATION

Manuscript Title:  Effects of Electrolytes on the Electrochemical Impedance Properties of NiPcMWCNTs Modified Glassy Carbon Electrode

Manuscript ID: nanomaterials-1713962  

We appreciate the reports on our manuscript. As a result of this, we submit a response to the comments on the manuscript for further consideration.

The reviewers' efforts at improving this manuscript are well appreciated. We have carefully considered the comments. The responses to all reviewers' comments are highlighted in yellow in the manuscript. 

Reviewer 2

  1. One might suspect that the NiPcMWCNTs-modified electrode is Ni(0)-nanoparticle and free-base phthalocyanine and MWCNTs deposited GC, not containing Ni-phthalocyanine complex. Usually, the preparation of metal phthalocyanine (M(II)Pc) is carried out by mixing with free-base phthalocyanine (H2Pc) with metal (II) species, such as Ni(OAc)2. I am unsure whether the metal introduction to the free-base phthalocyanine (H2Pc) occurs with Ni (0) nanoparticles or not. The formation of the Ni-phthalocyanine complex may be confirmed in the UV-Vis spectrum. However, the provided UV-Vis spectrum was indistinguishable because the spectrum presented may not be the spectrum in the solution.

Response: The EDX spectral has been included to indicate the presence of Nickel nanoparticles In the NiPcMWCNTs composites. See figure 3, page 7, line 192 – 199.

  1. Comment: If the electrode contains Ni (0)-nanoparticle, structural changes would occur in dilute aqueous sulfuric acid, which might participate in the characteristic feature of the electrode in dilute aqueous sulfuric acid. The stability experiment of the electrode in sulfuric acid (pH1) is recommended in the same manner in PBS (pH7).

Response: The stability of NiPcMWCNTs in the manuscript is not in ordinary PBS (pH7) but in 5 mM [Fe(CN)6]3-/4- solution produced in 0.1 M PBS at pH 7. Nonetheless, stability of NiPcMWCNTs in 0.1 M H2SO4 solution in 0.1 M KBrO3 (pH 1) at a scan rate 25 mVs-1 exhibited a current drop (9.4%). The Cyclic voltammogram is given below as requested. However, 0.1M KBrO3 solution in 0.1 M PBS (pH 7) gave no peak anode or cathodic peak.

  • Cyclic voltammograms of 20 scans of GCE-NiPcMWCNTs modified electrode in 1 M KBrO3 solution in 0.1 M H2SO4 (pH 1) at a scan rate 25 mVs-1

Reviewer’s remarks:

  1. Comment: line75: p => π?

Response: p has been replaced with π. Correction highlighted in yellow. See line 75, page 2.

  1. Comment: line 102: sulphuric => sulfuric

Response: Sulphuric has been replaced with sulfuric. Correction highlighted in yellow. See line 108, page 3.

  1. Comment: line 103: K[Fe(CN)6]3-/4- => [Fe(CN)6]3-/4-

Response: K[Fe(CN) 6]3-/4- has been replaced with K3-/4[Fe(CN)6]. Correction highlighted in yellow. See line 104, page 3.

  1. Comment: line 105: sodium hydrogen => sodium dihydrogen

Response: Sodium hydrogen has been replaced with sodium dihydrogen. Correction highlighted in yellow. See line 106, page 3.

  1. Comment: line 160: 101 => 100?

Response: 101 has been replaced with 100. Correction highlighted in yellow. See line 166, page 4

  1. Comment: line 205: Figure 3. Is the vertical axis absorbance?

Response: The vertical axis in Figure 3 is absorbance. See line 217, Now Figure 4.

  1. Comment: line 336: Figure 9. Is the vertical axis -Z”?

Response: The vertical axis is -Z”. Z” has been replaced with -Z”. Correction highlighted in yellow. See line 338, Now Figure 10.

  1. Comment: line 354: Figure 10, caption. 0.1 M H2SO4 (pH1) => 0.05 M or pH0.7?

Response: It is 0.1 M H2SO4 (pH 1) see the yellow colour, lines 379 – 380, Now Figure 11 (caption).

  1. Comment: line 356 and etc.: BrO3 => H3BO3 or NaBrO3?

Response: BrO3 has been replaced with KBrO3 in all pages it appears. Correction highlighted in yellow.

  1. Comment: line 439: NAB => SAB?

Response: NAB has been replaced with SAB. Correction highlighted in yellow. See line 464, 471, 478, 494 and in all pages it appears.

Round 2

Reviewer 2 Report

The manuscript has been improved. I think this manuscript will be acceptable for publication.